# A System-Dynamic Model for Human–Robot Interaction; Solving the Puzzle of Complex Interactions

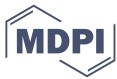

**Wouter Martinus Petrus Steijn *** , **Coen Van Gulijk, Dolf Van der Beek** and **Teun Sluijs**

Netherlands Organisation for Applied Scientific Research TNO, Sylviusweg 71, 2333 BE Leiden, The Netherlands
* Correspondence: wouter.steijn@tno.nl

**Abstract:** Cooperative robots in the workspace have an effect on safety that is not yet fully understood. This work collates pre-existing knowledge on human, technological and organizational factors for human-robot interaction and develops a system dynamics model that captures the complex interactions. Expert consultation in the form of a Delphi study is used to derive a tractable model from pre-existing puzzle pieces. A final model is presented, which contains 10 nodes and 20 relationships containing the three key outcome factors of human-robot interaction, viz. Safety, Efficiency and Sustainability. By combining these factors into a single tractable framework, this model bridges the gap between individual efforts from previous works in the field of robotics.

**Keywords:** human-robot interaction; system dynamics; Delphi study; OSH

## 1. Introduction

Cooperative robots, or cobots, operate in industrial workspaces. Unlike most industrial robots, which are fenced off and designed to work independently, these robots interact with humans in a shared working space. Consider automated guided vehicles (AGVs) that share the shop floor with staff to move goods between storage and shipment. Sharing the work floor introduces new scenarios for trips and collisions. Alternatively, consider a cobot arm on an assembly station. The operator and the cobot collaborate on the same task, share the same space, and interact almost as if both were humans. This close interaction between man and machine increases the efficiency of work but also creates new incident scenarios and poses new challenges to workspace health and safety because separation of man and machine, as dictated by traditional methods (and as prescribed by the European Directive 2006/42/EG machinery, annex 1) are no longer applicable. At the same time, the use of 'smart' technologies (either with or without AI) alone does not guarantee an injury-free environment. A recent investigation into warehouse safety indicates that employees in modern warehouses (with robots) are actually more at risk of injury [1]. So, technology alone is not the solution for safe human–robot interaction (HRI), and human and organizational factors should also be taken into account.

Take, for example, a robot arm on an assembly system with a workspace monitoring system that uses vision analytics to track the environment and identify humans approaching the robot arm. Evaluating the setup based solely on the number of incidents is not enough. The work process should ideally also be optimized and sustainable for human employees while keeping safety in mind. So aside from whether collisions are indeed prevented, the evaluation should also look into how often the workflow is interrupted due to employees entering its workspace during the course of a day or due to false identifications of employees approaching the workspace when, in fact, there are none. The evaluation should also take into account how employees experience the interactions. How does the robot communicate a warning signal to the employees entering its workspace? Is the reduction of speed sufficient for the employee not to experience any stress while being in its vicinity? A unifying framework to evaluate this setup is still lacking, as discussed in the next section.

A humans, technology, and organization (HTO) approach is proposed to better understand the complex interplay between employees and moving machines during HRI. In particular, proper organization of the HRI can be expected to have an even more direct impact on safety performance now employees are no longer physically removed from the machine. It requires a more dynamic kind of risk assessment that captures the HTO interplay much better than traditional methods can do, as the dynamics between two intelligent actors (a human and a cobot) mimic interactions traditionally only associated with human-human interactions.

This work proposes a System Dynamics (SD) approach as a method to capture the dynamic risk assessment for cobot collaborations. All the interactions that occur in complex phenomena of robot safety can be defined as 'wicked' problems [2]. To address wicked problems to the extent to which they might become manageable current research needs a holistic, integrated approach that is easily accessible enough to attract stakeholders to a more qualitative extent. The methodology should simultaneously be able to capture nonlinear behavior over time of this 'complex problem'. The method of System Dynamics is well-suited to address this complexity and is deemed appropriate to identify important leverage points prior to the real-life implementation, which could otherwise result in unsafe situations. The goal is to create a tractable model with the most important factors to model the underlying complexities for successful interaction that can subsequently be used as the basis for evaluation or risk management of existing HRI systems.

Successful interaction has been defined from an Occupational Health and Safety (OSH) perspective and therefore requires the interaction to be not only efficient but also safe and sustainable. Here, efficiency is defined as obtaining the desired outcome without wasting time and effort. Safety concerns the absence of adverse consequences on the user(s) and the environment [3,4]. Lastly, sustainability is defined as the goal that human–robot interactions should not make the human ill or disabled, either physically or mentally. This holds for both the short term as well as the long term.

### 1.1. Limitations of Safety Models for Working with Robots

Keeping the workplace safe is a complex problem in every industry. This complexity increases when robots are introduced as they bring additional uncertainties, with many determinants and small margins of error. Reducing these uncertainties within an increasingly complex system and making informed decisions on risk management requires more complex simulations compared to traditional methods. This all coincides with the complexity mindset and the use of computational modeling, such as System Dynamics (SD) methodology, to get a better understanding of the complex dynamics within (future) organizational systems. SD allows for a complex model of multiple factors and relationships and the dynamics that arise through interconnections that affect each other, as well as taking (pipeline) delays into account.

Until now, robot safety is generally treated in a fragmented way rather than providing the interplay between various relevant factors. Numerous papers provide knowledge on relevant factors that make human–robot interaction safer or more efficient. A majority of this work has focused on technological factors (see overviews [5–7]). Human factors received somewhat less attention (e.g., [8,9]), but some notable exceptions exist, such as the work by Neumann, Winkelhaus, Grosse and Glock [10], who developed a framework to assess the impact that new technologies have on human workers and overall efficiency and Baltrusch, Krause, de Vries, van Dijk and de Looze [11] who present various factors required to maintain good job quality.

Literature provides taxonomies or frameworks of relevant factors for the evaluation, classification or analysis of human–robot interaction (e.g., [12–15]). However, these papers do not fully address the full complexity needed to understand the intricacies of safety in human–robot interaction. What are the specific interplays in the various identified factors? How would intervening on one factor influence the other factors and the system as a whole?

These and other questions remain unaddressed by the majority of papers, whereas a more holistic modeling approach would provide insights into the possible answers.

One of the earliest attempts to deal with the complexity of safety (which is still in use today) is the Fine–Kinney method introduced in 1970 [16]. This method concerns a semi-quantitative approach for scenario-driven risk management of health and safety. There are few papers that try to link two or more factors together. Notable is the meta-analysis by Ötting, Masjutin, Steil and Maier [17], in which they investigate the relationship between the interface, controller and appearance of a robot with several indicators of success. The indicators they identified were performance, cooperation, satisfaction, acceptance, trust, and workload. Their results provide numerous important pointers for successful robot design.

Ideally, the findings and knowledge from various studies would be linked together to form a single SD model concerning HRI. In essence, many of the literature studies focused on retrospective data, whereas this research aims to gear toward prospective behavior through retrospective patterns. In other words, existing detailed studies provide individual puzzle pieces (see Figure 1), which, when combined in an SD model, provide a holistic overview of human–robot interaction. Most studies only produce one node-relationship-node combination, and a method was needed to piece different relationships together. The process of piecing different parts together is best illustrated as a puzzle where different parts come together to create an overview.

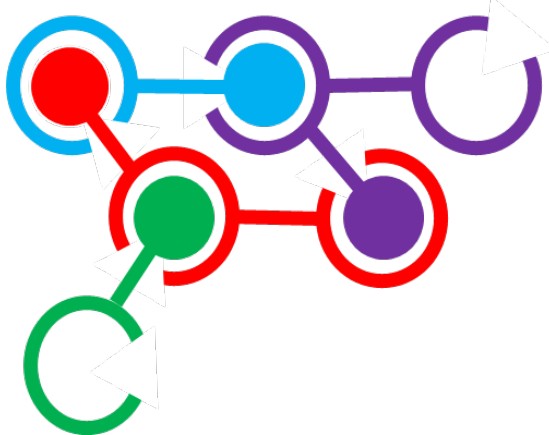

**Figure 1.** IQ link puzzle game: a metaphor for bringing literature evidence together in a single model.

SD is specifically well-suited to address the described situation. Most of the data in the literature are provided on an aggregated level and are still rather limited, so the research and resulting model ought to be shaped in a broader perspective to address these datasets and still hold the possibility to extend the knowledge from limited data with a more qualitative approach. In general, there are many delays and fuzzy behavior that occur on the human side in relation to robot safety that we aim to capture. All these interactions result in the choice of System Dynamics (see also Figure 1 in [18]).

*1.2. Concatenation of Relationships to Create an SD Safety Model*

The concatenation of relationships reported in the literature can be done in various ways. To explain this IQ-Link puzzle (as illustrated in Figure 1) is used as a metaphor. If you consider that individual robot safety papers link just two (and sometimes three) factors, an individual paper adds a single piece of the puzzle (e.g., the green piece or the red piece). With sufficient papers, influence factors appear in multiple papers linking more and more factors (i.e., there are many different pieces that connect factors). If there is not too much controversy, a network of influence factors can be constructed just from the evidence found in the literature (a bit like that shown in Figure 1). However, it would still be necessary to assess whether the puzzle was put together in the right way and separate major influence factors from minor ones, which was done with expert elicitation. Fundamentally, this work reports on the process and findings of the expert consultation process.

The majority of the work for the first step (collecting pieces of the puzzle) was reported in a paper on 'optimal model HRI' from 2020 [7,19,20]. That work gathered a large number of relevant factors that affect HRI, distinguishing human, robot and environmental (including organizational) factors as part of the overall system (thereby following the approach from Reason [21]) and classified them in a matrix called based on systems approach using inputs, manipulations and outputs. That work added further distinction between factors affecting physical and technical characteristics, knowledge and underlying processes or procedures, attitude, and experience. This classification method differs from the one followed in this paper in the way that the pieces of the puzzle were classified in bins (the equivalent being putting puzzle pieces of the same color together in bins). The shortcoming of this approach was that even if a classification system that categorized different groups of influences increased our understanding of safe human–robot interactions, creating a holistic model to model interactions remained elusive.

Concatenation based purely on literature sources was reported in 2022 [22]. This was the first step in transforming a matrix model into a system dynamics model by extracting the relationships between the identified factors from the literature. This was done using the IQ-link metaphor in Figure 1 and combining the knowledge from individual nodes (factors) together into a comprehensive and coherent model. The resulting model, an overview of the 'atomic' studies, weaved into a single network. The resulting model has 25 factors and 40 relationships (see Table 1). The factors in the model describe the features or functions of the robot, task or employee involved in the HRI. How this function or feature is present is not per se relevant. For example, communication between the robot and the employee can take various forms, from visual to auditory cues. When evaluating the robot, the important thing is whether the communication is successful (i.e., a message is transferred from the robot to the employee or vice versa) but not in what form the communication takes place (e.g., through bleeps, light signals or verbal commands).

This work reports how the model was subsequently validated and improved through expert consultation and how it led to several major and minor changes to end up with a practical and tractable SD model. The elicitation process was performed as a Delphi study and is explained in the next section.

## 2. Method

The method for elicitation was a Delphi panel. A Delphi study is frequently used to develop consensus on what a specific new concept or development means, especially when opinions are expected to be diverse and to define what factors are relevant for addressing a new complex issue or what factors should be regarded as relevant for future developments. Delphi is a widely used tool for measuring and aiding forecasting and decision-making in a variety of disciplines by eliciting and combining expert judgments. A Delphi study generally consists of several phases, i.e., rounds [23–25] and provides a systematic methodology to collect the opinions of a small but knowledgeable sample of experts to arrive at a meaningful consensus. In using the Delphi technique, one controls the exchange of information between anonymous panelists over a number of rounds (iterations), taking the average of the estimates on the final round as the group judgment [26–28]. Classical Delphi studies consist of four rounds. However, two rounds can be sufficient as well [23,29].

Approximately 100 experts with varying backgrounds related to human–robot cooperation were invited to participate in this study, including scholars of selected articles in our prior literature exploration not stemming from the personal networks of the authors, thereby significantly increasing the accuracy of the method and maximizing diversity in the group (preventing group-think or other unfavorable group dynamics). The sample also included practitioners from the field primarily derived from Dutch normalization and standardization work groups on robotics. Many experts were located in the Netherlands, but there were contributions from European OSH institutes and international OSH institutes in Japan and the USA. The Delphi study consisted of three rounds: two survey rounds and a live panel session. Seventeen experts participated in the first round, thirteen experts participated

in the second round, and eight experts participated in the panel session, one of which had not participated in the previous rounds (see Table A1 in Appendix A for an overview).

*Design of Three Rounds*

In the first two survey rounds, experts were asked to rate the factors and relationships that were currently in the model and were given the opportunity to suggest factors and relationships that were missing in the model (see Table A2 in Appendix B). Experts were given an abridged version of the definitions for the factors (see Appendix C).

The third round was an online panel session using Mural, which is an online software tool that allows large groups to collaborate simultaneously. Here, the model, with alterations based on the survey rounds, was discussed. Experts were given the opportunity to provide feedback on the factors and relationships in the model during a silent brainstorm. During a silent brainstorm, experts were able to provide their comments in Mural without the interference of the other experts. In the following discussion based on these comments, experts were also asked to what degree they considered the model suitable for quantification of the complex interplay between the HRI factors and to what extent the model is applicable for human–robot cooperation in the industrial context. Two experts participated through a separate session as it was impossible to find a timeslot in which all (international) experts were able to participate (the time difference between Japan and the US (east coast) USA is 13 h).

## 3. Results

### 3.1. Survey Rounds

Rounds 1 and 2 of the Delphi panel were survey rounds. In this survey, experts were able to evaluate the factors and relationships in the model and propose new factors and relationships they deemed missing. Below we will describe the results for each of these elements in turn.

### 3.1.1. Evaluation of Factors in the Model

In round 1, experts were asked to provide the five factors they considered most and least relevant. In order to determine the most and least relevant factors, a cut-off was chosen to only consider factors that were ranked in the top 5 by 30% (5 or more) of the experts. Next, in round 2, they were given the opportunity to reflect on the factors that were chosen most often as most or least relevant.

Concerning the least relevant factors, six factors met the criteria in round 1: *Appearance* (Selected: 13), *Post-collision measures* (Selected: 9), *Proximity* (Selected: 7), *Vigilance* (Selected: 5), *Trust* (Selected: 5), and *Speed* (Selected: 5). In round 2, experts were given these factors that would be considered for exclusion of the model. In response, experts argued that while these factors may have been selected as least relevant, they still hold relevance. However, specifically for the factors of *Trust* and *Speed*, several experts argued that these should not be removed from the model. As a result, the factors of *Appearance*, *Post collision measures*, *Proximity* and *Vigilance* were considered eligible for removal from the model.

Concerning the most relevant factors, seven factors met the selection criteria in round 1: *Interaction design* (Selected: 12), *Safe by design* (Selected: 13), *Reliability* (Selected: 7), *Transparency* (Selected: 6), *Training* (Selected: 5), *Communication* (Selected: 5) and *Human error* (Selected: 5). In round 2, experts were given these factors for consideration. Experts did not show any disagreement with the factors in the list, providing arguments for their importance.

### 3.1.2. Evaluation of Relationships in the Model

Considering the first criterion, we find that for 29 relationships, 14 or more experts (80% or more) agreed that the relationships were true. This criterion was not met for any relationship regarding a relationship being non-existent or doubtful. In addition to the first criterion, two additional criteria were set that had to be met before considering consensus to exist. Only in one case did these criteria affect one of the 29 relationships that met the first

criterion. For the relationship *Fault avoidance affects Reliability*, eight experts indicated the relationship as possibly true, crossing the threshold of 40% (or seven experts in the sample). As a result, experts in the sample were considered in consensus with 28 relationships from the model, leaving 12 relationships for consideration in round 2.

In round 2, the experts together made 59 changes in their answers from a total of 156 possible changes (38%). For one relationship, these changes meant that it now meets the criteria for consensus. Three other relationships only met 2 out of 3 criteria set for consensus. See Table 1 for an overview. Overall, the changes in response had little impact on the overall evaluation of a relationship.

Nine relationships (in italics in Table 1) were removed from the model as one factor involved was selected as least relevant and opted for removal (see Section 3.1). This concerns four relationships for which consensus was met and five relationships for which no consensus was met. This leaves 25 relationships for which consensus was met that will be included in the model. The remaining six relationships for which no consensus was met (concerning either their relevance or irrelevance) will also be included but with a lower weight given to them to reflect the uncertainty demonstrated by the panel.

**Table 1.** Consensus results and decisions for relationships in the model.

| ID | Parent | Child | Round 1 Consensus | Round 2 Consensus | Decision |
|---|---|---|---|---|---|
| E1 | Fault avoidance | Cognitive workload | - | O | Retained |
| E2 | Communication | Cognitive workload | X | | |
| E3 | Stress | Cognitive workload | X | | |
| E4 | Fatigue | Cognitive workload | X | | |
| E5 | Trust | Complacency | X | | |
| E6 | Cognitive workload | Complacency | - | - | Retained |
| E7 | Reliability | Complacency | X | | |
| E8 | Pre-collision measures | Efficiency | X | | |
| E9 | Coordination | Efficiency | X | | |
| E10 | Human Error | Efficiency | X | | |
| E11 | Interaction design | Efficiency | X | | |
| E12 | Transparency | Efficiency | X | | |
| E13 | Proximity | Efficiency | - | - | Removed |
| E14 | Situational awareness | Human error | X | | |
| E15 | Interaction Design | Human error | X | | |
| E16 | Appearance | Job Quality | - | - | Removed |
| E17 | Pre-collision measures | Job Quality | - | O | Retained |
| E18 | Transparency | Job Quality | X | | |
| E19 | Proximity | Job Quality | - | | Removed |
| E20 | Directability | Reliability | - | - | Retained |
| E21 | Fault avoidance | Reliability | O | O | Retained |
| E22 * | Post-collision measures | Safety | X | | Removed |
| E23 | Fault avoidance | Safety | X | | |
| E24 | Human Error | Safety | X | | |
| E25 | Transparency | Safety | X | | |
| E26 | Pre-collision measures | Safety | X | | |
| E27 | Safe by design | Safety | X | | |
| E28 | Situational awareness | Safety | X | | |
| E29 | Communication | Situational awareness | X | | |
| E30 | Complacency | Situational awareness | - | - | Retained |
| E31 | Training | Situational awareness | X | | |
| E32 * | Vigilance | Situational awareness | X | | Removed |
| E33 | Proximity | Stress | - | - | Removed |
| E34 | Transparency | Stress | - | X | |
| E35 | Speed | Stress | X | | |
| E36 | Appearance | Stress | - | - | Removed |
| E37 | Communication | Transparency | X | | |
| E38 | Reliability | Trust | X | | |
| E39 * | Cognitive workload | Vigilance | X | | Removed |
| E40 * | Fatigue | Vigilance | X | | Removed |

Note. X indicates the relationships for which consensus was met for all three criteria. O indicates the relationships for which 2 criteria were met, but a larger proportion than 40% had selected 'possibly true'. Some relationships were removed despite consensus on its relevance, as one involved factor was considered eligible to be removed (see Section 3.1.1). These are marked with an asterisk.

### 3.1.3. Proposed Factors

In round 1, the criteria were set to consider new factors for inclusion that were 'proposed by more than two experts independently'. Experts proposed a great number of factors, but most factors were only suggested by a single expert or hinted at factors that were already included. Ultimately, four new factors were extracted from the input that were presented to experts in the second round:

- *Outside interference*: intentional and unintentional outside interference by hackers or viruses;
- *Output demands*: procedures and rules in place concerning workspace and output demands;
- *Technology Acceptance*: to what degree does the operator accept the (need to) use robots and their usefulness;
- *Experience* (with robots): how much experience does the operator have with operating (similar) robots.

In round 2, experts were asked to comment on these factors and indicate the relevance of each factor. For inclusion in the model, we had set the criteria to have at least 80% (i.e., 11 or more) experts indicate the factor as relevant and have no more than 10% (i.e., 1 or none) experts indicate the factor as not relevant. This was the case for *Output demands* and *Technology acceptance*, with all experts indicating these as relevant, of which nine considered them very relevant. Experience also met the criteria but with nine experts considering it somewhat relevant and one expert indicating it not to be relevant.

In the comments, experts indicated that *Outside interference*, although relevant, should not be included in the scope of this model and should be considered a separate issue to be solved independently. Concerning *Experience*, which did meet the criteria to be included, some experts noted that it had some overlap with factors such as *Training* or *Technology acceptance*.

### 3.1.4. Proposed Relationships

In round 1, the criteria to consider new relationships for inclusion were set so that they were 'proposed by more than two experts independently'. Experts proposed a total of 25 new relationships for inclusion in the model. Although few to none were suggested by more than one expert, the authors decided to present all relationships to the experts in the second round and ask them to indicate the relevance of each relationship.

Some of these relationships had been slightly rephrased to match the factors already existing in the model. For inclusion in the model, the criteria were set to have at least 80% (i.e., 11 or more) experts indicate the factor as relevant and have no more than 10% (i.e., 1 or none) experts indicate the factor as not relevant. In Table 2, all relationships are presented and indicated for which relationships the criteria were met. The criteria were met for 15 relationships, which will be included in the model. As none of the new relationships were related to the newly proposed factor Experience, it was decided not to include this factor in the model as a separate entity.

**Table 2.** Reported relevance in round 2 of relationships proposed in round 1.

| ID | Parent | Child | Round 2 Consensus | Decision |
|----|--------|-------|-------------------|----------|
| N1 | Interaction design | Cognitive workload | X | Included |
| N2 | Interaction design | Directability | X | Included |
| N3 | Directability | Efficiency | | |
| N4 | Speed | Efficiency | X | Included |
| N5 | Output demands | Efficiency | X | Included |
| N6 | Safe by design | Efficiency | X | Included |
| N7 | Fault avoidance | Efficiency | | |
| N8 | Training | Experience | | |

**Table 2.** *Cont.*

| ID | Parent | Child | Round 2 Consensus | Decision |
|---|---|---|---|---|
| N9 | Interaction design | Fatigue | | |
| N10 | Stress | Fatigue | X | Included |
| N11 | Technology acceptance | Job quality | X | Included |
| N12 | Directability | Job quality | X | Included |
| N13 | Output demands | Job quality | | |
| N14 | Safe by design | Job quality | | |
| N15 | Output demands | Safety | | |
| N16 | Interaction design | Situational awareness | X | Included |
| N17 | Output demands | Stress | X | Included |
| N18 | Cognitive workload | Stress | X | Included |
| N19 | Technology acceptance | Stress | X | Included |
| N20 | Safe by design | Stress | | |
| N21 | Fault avoidance | Stress | X | Included |
| N22 | Pre-collision measures | Stress | X | Included |
| N23 | Safety | Stress | | |
| N24 | Job quality | Trust | | |
| N25 | Interaction design | Trust | X | Included |

### *3.2. Panel Session*

The main concerns and remarks of the experts will be briefly summarized below. Several main themes could be identified which emerged from the discussion: *specific remarks on relationships and the model, the complexity of the model, the scope of the model and challenges concerning the quantification of factors in the model.*

### 3.2.1. Specific Remarks on Relationships

During the so-called silent storm, panelists were able to provide specific remarks on all relationships in the model. In addition, panelists were able to vote on each remark allowing us to identify the more important comments. An important consideration is that it appears that it is unlikely that complete consensus will be achieved as numerous suggestions for new or adapted existing relationships were made, which are to be taken into consideration in the next steps in the development of the model.

A specific issue was that panelists sometimes took issue with the name of a specific factor as it may suggest covering more than originally intended. For example, job quality in the current model was only limited to 'interaction quality' as it did not, for example, include satisfaction with payment. The exact naming and definition of each factor will need to get extra care given the complexity of the overall model.

### 3.2.2. Specific Remarks on the Model

A specific remark was centered around the question of why a dynamic system model was used. This remark was closely related to questions on the exact purpose of the model. The original intended purpose was risk management and evaluation. However, the expert's proposed uses of the model for real-time monitoring of a cobot application or a model to support the design of safe robots would have required different choices throughout the development process. Several alternatives to SD modeling were proposed, including a concentric model with safety at the core and different layers of factors surrounding it or simplifying to an input-process-output model with clearly defined moderators.

Another aspect was that the various factors were not properly identified or ordered in the model. For example, both static (e.g., pre-collision measures) and dynamic (e.g., trust) factors were included in the model without an explicit distinction. Static factors need to be established once and, from that point on, have a fixed influence on the model, whereas dynamic factors can be remeasured over time and are likely targets for interventions to enact change in the model. Better distinction between such factors would therefore improve the model.

### 3.2.3. Complexity

All experts agreed that the proposed model was extensive. They were, however, divided in the degree to which this complexity was necessary given the modeled topic or that it only led to unnecessary chaos. One element that contributed to this complexity was the fact that various relationships appeared to have been duplicated in the model through both direct and indirect pathways.

Several suggestions were made to simplify the model. One suggestion was to split the model into multiple models: for example, a model that deals with human factors and a model that deals with (technical) safety. However, precisely the inclusion of such varying factors was also seen as a strength of the current model. Another suggestion was to cluster various factors to make the model better readable.

### 3.2.4. Scope

The experts warned that the resulting model has a Western–European bias. It is indeed true that the model has been developed from a western European perspective, and the panel members were mostly from western European countries. As such, the model may not always fully reflect common practices in, for example, America or Eastern Europe. For example, the exclusion of post-collision measures from the model was considered an example of Western European thinking.

### 3.2.5. Quantification

The experts identified the quantification of the various factors as an important challenge. For each factor, a relevant scale and measurement method need to be decided on, and the relationship between these factors (e.g., linear or quadratic). A complicating factor for this is the fact that not all factors are on the same level (e.g., safe by design, which is based on the requirements of the machine directive compared to the fatigue of the operator), which may result in parameters that are difficult to compare.

### 3.3. Concatenation of Factors

This step aimed to reduce the complexity of the model while maintaining all elements that had been identified as relevant. In order to do so, it was decided to cluster several factors together that did not violate the input of experts whilst creating a comprehensive set of clusters. These clusters were divided into human, technological and organizational clusters. Table 3 provides an overview of the defined clusters and which original factors fall under each cluster.

**Table 3.** Clusters in the final model.

| Type | Clusters | Original Factors | |
|---|---|---|---|
| Human | Human vigilance | Situational awareness; Fatigue; Complacency; Cognitive workload; Human error; Training | The capacity for sustained attention. |
| Human | Human attitude | Trust; technology acceptance; Training | The beliefs the operator has concerning the system. |
| Technological | Safe design | Pre-collision measures; safe by design | Technical design that makes the system as safe as possible. |
| Technological | Machine reliability | Fault avoidance; reliability | The degree to which the system provides correct service that can justifiably be trusted. |
| Technological | Ergonomics | Interaction design; Communication | Design properties that facilitate the human–robot interaction. |
| Technological | Human-in-control (principles) | Transparency; Directability | The measure of (perceived) control the operator has over the system. |

**Table 3.** *Cont.*

| Type | Clusters | Original Factors | |
|---|---|---|---|
| Organizational | Task design | Speed; Coordination; Output demands | Task properties related to the human–robot interaction. |
| Output | Safety | Safety | The absence of harm and/or damage to the user(s) and the environment. |
| Output | Efficiency | Efficiency | Obtaining the desired outcome without wasted time, effort and resources. |
| Output | Sustainability | Stress; Job quality | Actual and perceived (psychological) discomfort for operator while interacting with the robot. |

Based on the relationships between the factors with the clusters, the relationships between clusters were established (see Table 4 for an overview). This resulted in a model with 10 clusters and 22 relationships.

**Table 4.** Relationships in the final model.

| Parent | Child | Original Relationships (see Tables 1 and 2) | Label |
|---|---|---|---|
| Ergonomics * | Efficiency * | E11; E15 | Facilitation |
| Human-in-control | Efficiency | E12 | Coordination |
| Safety | Efficiency | E8; E10; E26; N6 | Interruptions |
| Task design | Efficiency | E9; N4; N5 | Optimization |
| Ergonomics | Human attitude | N25 | Ease of use |
| Machine reliability | Human attitude | E38 | Trust |
| Ergonomics | Human-in-control | E37; N2 | Facilitation |
| Ergonomics | Human vigilance | E2; E15; E29 N1; N16 | Facilitation |
| Human attitude | Human vigilance | E5 | Complacency |
| Machine reliability * | Human vigilance * | E1; E7; E21 | Dependability |
| Sustainability | Human vigilance | E3; N10 | Stress |
| Human-in-control | Machine reliability | E20 | Responsiveness |
| Human-in-control | Safety | E25 | Predictable |
| Human vigilance | Safety | E24; E28 | Situational awareness |
| Machine reliability | Safety | E23 | Fault avoidance |
| Safe design | Safety | E27 | Safe |
| Human attitude | Sustainability | N11; N19 | Acceptance |
| Human in control | Sustainability | E18; E34; N12 | (Perceived) control |
| Human vigilance | Sustainability | E4; N18 | Workload |
| Machine reliability | Sustainability | N21 | Dependability |
| Safety | Sustainability | E17; N22 | Anxiety |
| Task design | Sustainability | E35; N17 | Pacing |

Note. The relationships E6; E14; E30; E31 are missing in the table above. These relationships refer to reinforcing loops within the Human vigilance cluster. The relationships *Ergonomics—Efficiency*, and *Machine reliability—Human vigilance*, marked with an asterisk, were removed from the model to remove some redundancy.

In order to remove some redundancy from the model, the relationships *Ergonomics—Efficiency*, and *Machine reliability—Human vigilance* (marked in Table 4) were removed from the model. The former relationship already exists in the route through *Human-in-control*, i.e., ergonomics improve efficiency by making the robot or machine better directable (a human-in-control principle). The latter existed in the route through *Human attitude*, i.e., the dependability as a result of a more reliable robot or machine increases trust, which in turn will lower the vigilance. By taking these modifications in mind, the resulting model is shown below (Figure 2).

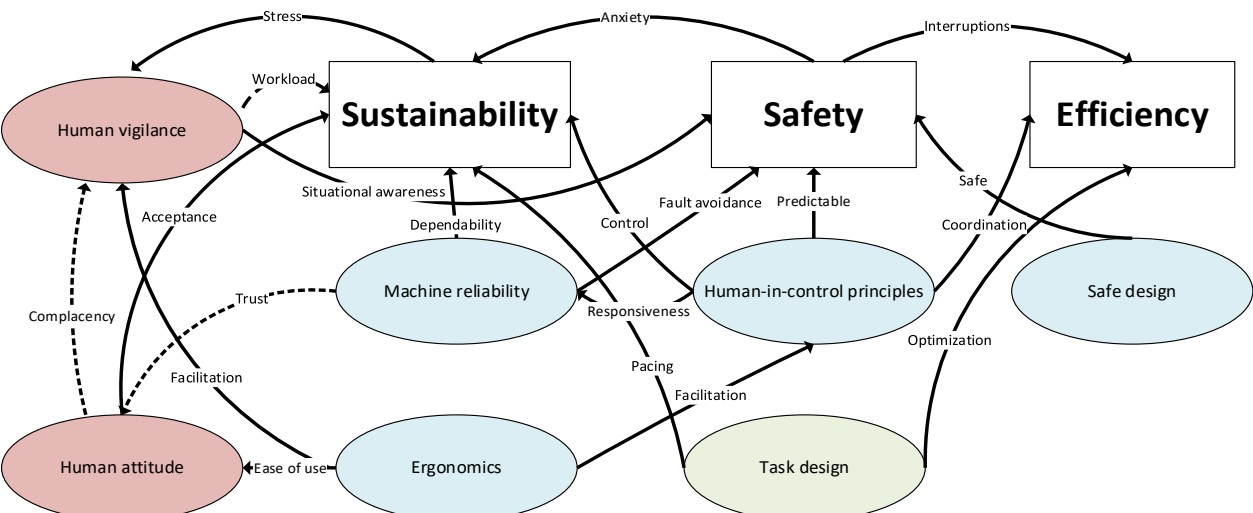

**Figure 2.** Visualization of the clusters and relationships of the final model, as listed in Table 4. Red concerns human clusters, blue are technological clusters, green is the organizational cluster, and white are the output clusters.

## 4. Discussion

This paper introduces a tractable model for human–robot interaction in the industrial context. The goal was to validate and improve an earlier, more complex version of this model through expert consultation in a Delphi panel. This model contains the features and functions that an HRI system should contain. It abstains from prescribing or describing technologies to fulfill these functions or features. As such, this model could be considered technology neutral. Future and existing technologies could fulfill the functions and features. Whether they do or do not, they should be evaluated as part of workplace risk management. This model states what functions and features should be taken into account for that assessment. Note the inclusion of the most advanced technologies does not suffice for safe HRI. There are examples shown [1] where the presence of the latest technology increased injury rates among the workforce in modern warehouses.

Valuable input was collected from international experts in the field of human–robot interaction and implemented their feedback on the model. During this process, the original model consisting of 25 factors and 40 relationships to a final SD model was bought back to 10 clusters and 20 relationships. Note that the clusters still encompass the original factors and relationships (as shown in Table 3) but allow for a more comprehensible overview. In addition, consensus was reached regarding the relevance of the majority of the included clusters and relationships included in this model.

The presented model proves to be effective in reducing a myriad of factors and interrelationships associated with HRI into a tractable framework, containing each element of HTO without losing the intent. It provides a springboard for the development of operational methods, for instance, for a design-aid quantification using standard SD quantification techniques or workplace evaluation techniques using questionnaires. Ultimately, a practical and comprehensive HRI risk assessment method is envisaged for the introduction of interactions between human and non-human intelligent actors, potentially extending that to a monitoring system. The next step will be to further define and quantify the interrelationships of the clusters. A particular strength of the current model is that all clusters are of a similar abstraction level.

The main challenge will lie in the operationalization of the clusters (and the underlying factors) for this purpose. This is closely related to the quantification of the model. The quantification of an SD model requires a fundamentally different research approach than the design of the model, which is why it isn't reported in this paper. The authors believe the model will help reevaluate how HRI systems can be monitored or evaluated. The

method should aim at quantifying each of the ten clusters identified in the model. The exact method to approach this is not set in stone. For example, there are numerous ways in which human attitude or task design can be operationalized. As said earlier, that work differs fundamentally from work reported here. Despite that, this work gives several pointers on how each cluster can be operationalized.

Take, for example, the cluster *Safe design*. Table 3 shows that this cluster contains the factors of *Pre-collision measures* and *safe by design* Pre-collision measures could be determined by comparing the HRI system under scrutiny against several pre-defined categories where each category has a defined effect. When it is missing, it is bad, while the inclusion of multiple technological solutions to address it is better (e.g., through robot vision potentially supported with AI capable of predicting human behavior). The factor *safe by design* could be operationalized based on safe by design principles. This may be as simple as eliminating sharp edges or using more lightweight materials or be more complex such as limiting a variety of parameters such as the velocity and potential of force extortion [5,6,30]. Taken together, this could give a score on the cluster *Safe design*.

This paper does not address the issue of operationalization, as it constitutes a fundamentally different research approach. For example, within a cluster, some elements might be more relevant than others, but that would require a separate study into the factors within that cluster. Furthermore, how each cluster is operationalized will define which methods can be used to collect data. Consider, for example, human attitude. Relevant data could be gathered through surveys, interviews or physical measurements (e.g., skin-conductance measurements). But equally, other clusters might be estimated through hard variables. Hard variables are data streams that are generated, for example, by a workspace monitoring system. Consider, for example, production speed, the number of interruptions, or the movement speed of the robot.

It should be noted that it is very unlikely to obtain a complete consensus on all factors and relationships that should be included in a complete model. Closely related is the fact that many relevant factors for human–robot interaction exist under many different guises and names with sometimes few or minor differences. For example, in scientific papers, one may encounter the dependability of a robot, which as a term, is missing in this model. Dependability is an umbrella term covering, among other things, the safety and reliability of the robot. Both of these factors are included in the model. As a result, to avoid redundancy by including various factors under all their guises, sometimes decisions had to be made for a specific definition. As a result, the model may always be under contention as some specific terms will not be included but may have been included in some other form. This emphasizes the need that all elements in the model will receive exact definitions and descriptions to formulate the intent and scope.

The panel sessions uncovered an interesting paradox. On the one hand, experts seemed to agree on the comprehensive nature of the model bordering very close to being too complex. On the other hand, experts proposed many new relationships that they deemed missing, while no relationships were marked conclusively as being irrelevant (or less relevant) for the model (although some relationships were considered superfluous as direct and indirect relationships overlapped). This shows an interesting challenge to balance the model between, on the one hand, the attempt to be complete and includes all relevant factors and relationships while, on the other hand keeping the model interpretable and readable. When domain experts consider a model to be complex, this will be amplified when presented to individuals with no specialization in the HRI domain. It also illustrates the added value of the methodology followed here. Individual experts will struggle to make informed decisions. Gathering multiple experts in a panel allows informed decision-making based on objective criteria of group consensus.

The resulting model is straightforward. The model does not contain all factors and relationships that are relevant to ensure safe HRI. This was never the goal. Instead, this model narrows down to the most important factors and relationships that should be taken into consideration for risk analysis of an HRI within a collaborative workspace. The factors

included still represent the human, machine and environment (organization). Other work also emphasizes addressing these three elements. Take, for example, the white paper on safety in the future [31], where it is argued that safety should be addressed from the tripartite system: human, machine and environment.

The HTO framework apparently remained relevant and necessary for addressing safety two decades since its original introduction [21], and serious attempts at risk management should address all three pillars. This does mean that a legal framework, such as the machine directive, by itself will fall short of guaranteeing safe HRI applications. This illustrates that the full field of occupational safety expertise is not addressed by just the legal framework; the machine directive will never set rules on human attitudes or vigilance, yet these human factors need to be addressed to optimize HRI. Nor will a safe, cobot and motivated workforce guarantee safe HRI when the task design is focused solely on productivity. Only by addressing all three aspects will HRI be optimized.

Although the presented model is an important step forward toward effective risk management of HRI applications, some limitations should be noted. First, as participating experts remarked, the model is primarily based on Western European notions of safety and sustainability. These criteria may differ in different regions of the world and could lead to different prioritization of factors (i.e., focus on post-collision measures instead of pre-collision measures). Second, the work primarily focuses on HRI applications within the industrial context. As a result, this model might be less appropriate for risk management of HRI applications in other sectors (such as the catering industry and healthcare).

## 5. Conclusions

This work presents a human–robot interaction model that has aggregated relevant factors to a tractable level. The model, as it is now, could be used as a supportive tool when discussing the cobot safety of an existing or planned application. In addition, it can be used to clarify, for example, why the machine directive will not suffice to make cobot applications safe and sustainable. The next step will be to make the model predictive. Could this model, once quantified, help demonstrate what the consequences are if a certain cluster is mismanaged or what the impact of specific intervention might be? The current model is ready for quantification into a data model, but that requires fundamentally different research from this work, mostly focusing on quantification and not on consensus gathering. In any case, it was demonstrated that it is possible to combine the many puzzle pieces that are available in the scientific literature and combine them into a single comprehensible and sufficient model. By doing so, the next step was taken towards the development of a digital safety monitoring tool for cobot applications.

**Author Contributions:** Conceptualization, W.M.P.S., C.V.G. and D.V.d.B.; methodology, W.M.P.S. and T.S.; formal analysis, W.M.P.S.; writing—original draft preparation, W.M.P.S.; writing—review and editing, C.V.G.; funding acquisition, D.V.d.B. All authors have read and agreed to the published version of the manuscript.

**Funding:** This research received no external funding.

**Institutional Review Board Statement:** The study was conducted in accordance with the Declaration of Helsinki and approved by the Institutional Review Board (or Ethics Committee) of TNO (07-09-2021).

**Informed Consent Statement:** Informed consent was obtained from all subjects involved in the study.

**Data Availability Statement:** The data are not publicly available due to safeguarding the privacy of the participants in accordance with Declaration of Helsinki.

**Acknowledgments:** This research has been made possible thanks to the support of the Dutch Ministry of Social Affairs and Employment regarding the TNO Occupational Safety Research Program 2021.

**Conflicts of Interest:** The authors declare no conflict of interest.

## Appendix A

**Table A1.** Expertise typology provided by the expert.

| | | Survey Round 1 | Survey Round 2 | Panel |
|---|---|---|---|---|
| 1 | human factors, ergonomics | x | | |
| 2 | physical robot interaction, artificial intelligence | x | | |
| 3 | human factors, interaction design, visual and tactile perception, cognitive psychology | x | | |
| 4 | human factors; interaction design | x | x | |
| 5 | human-machine interaction, human–robot interaction, mobile robots, mechatronics, development of robots equipped with arms, human factors, safety systems for safe human–robot interaction, technology acceptance, usability | x | x | |
| 6 | human factors and ergonomics specialist with experience in robotization and human-robot collaboration | x | x | x |
| 7 | working in the field of testing and certification of packaging and food machinery. There, more and more robots and cobots are used.considering physical contact, safety of the control system and further safety measures according to Machinery Directive | x | x | |
| 8 | project leader on OSH in a warehouse with robot systems, moderator of robotics workshops | x | x | |
| 9 | futurist risk expert, organizational risk adviser | x | x | x |
| 10 | social robots, verbal and non-verbal human–robot interaction, human factors, artificial intelligence, human–robot teaming | x | | |
| 11 | human factors, ergonomics | x | x | x |
| 12 | mechanical engineering, industrial design, human factors | x | x | x |
| 13 | human factors, organizational psychology | x | x | x |
| 14 | robotics, collaborative robots, physical robot interaction, automation, operational safety | x | x | x |
| 15 | human factors and ergonomics | x | x | |
| 16 | human factors, cognitive systems engineering | x | x | |
| 17 | human factors engineer for human technology interaction | x | x | |
| 18 | - * | | | x |

\* Experts were not required to leave a typology.

## Appendix B

**Table A2.** Consensus criteria.

| | Round 1 | | Round 2 | |
|---|---|---|---|---|
| | **Method** | **Consensus** | **Method** | **Consensus** |
| Existing factor | Select 5 factors least essential | Selected by at least 30% of experts | Chance to respond to factors opted for removal. | More than one expert arguing against removal. |
| Existing relationship | Rate on a 6-point scale * | - At least 80% selecting option 1, 2 or 3, or 4, 5, or 6<br>- Fewer than 40% selecting option 3 or 4.<br>- No more than 10% selecting the extreme opposite answer (1 or 6) | Reconsider answers for relationships for which no consensus was met. | See round 1. |
| New factor | Suggest new factors | Suggested by more than one expert | Rate new factors on three-point scale ** | - At least 80% selecting option 2 or 3, or 1.<br>- No more than 10% selecting the extreme opposite answer (1 or 3). |
| New relationship | Suggest new relationships | Suggested by more than one expert | Rate new factors on three-point scale ** | - At least 80% selecting option 2 or 3, or 1.<br>- No more than 10% selecting the extreme opposite answer (1 or 3). |

\* 1—Improbable, 2—Doubtful, 3—Difficult to determine, 4—Possibly true, 5—Probably true, 6—Confirmed by independent source. ** 1—Not relevant, 2—Somewhat relevant, 3—Very relevant.

## Appendix C

Safety is defined as the absence of harm and/or damage to the user(s) and the environment.

Efficiency refers to obtaining the desired outcome without wasted time, resources and effort.

Job quality refers to how the operator perceives working with the robot.

Stress refers to the (psychological) discomfort the operator experiences while interacting with the robot.

Appearance refers to the physical appearance of the robot but also its behavior.

Cognitive workload refers to the mental effort an operator endures during a task.

Communication refers to the transfer of information or data from the robot to the operator.

Complacency concerns overreliance on the system or automation by the operator.

Coordination concerns task allocation during the interaction.

Directability concerns the measure of control the operator has over the robot.

Fatigue is a mental state of (extreme) tiredness or a lack of energy.

Fault avoidance refers to the ability of the robot to avoid succumbing to faults.

Human error are slips, lapses and mistakes of the operator while performing his or her task, or unintentional errors.

Interaction design refers to the quality of the interfaces and/or controls through which the operator interacts with the robot.

Post-collision measures refer to all (technical) methods that are implemented to mitigate harm after a collision between the robot and the operator.

Pre-collision measures refer to all (technical) methods that are implemented to avoid a collision between the robot and the operator.

Proximity refers to the distance between the operator and the robot.

Reliability is the continuity of correct service that can justifiably be trusted.

Safe by design refers to all steps taken in the design phase to make the robot inherently safe.

Situational awareness concerns the person's mental model of the world around them.

Speed concerns the velocity with which the robot moves during the interaction.

Training is the acquisition of skills or knowledge to undertake a specific task.

Transparency refers to the clarity of the operator on what a robot does, what it will do next and why.

Trust refers to the firm's belief that the system is reliable.

Vigilance refers to a capacity for sustained effective attention when monitoring a situation or displays for critical signals, conditions or events to which the observer must respond.

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
