# Peer review of "A System-Dynamic Model for Human–Robot Interaction; Solving the Puzzle of Complex Interactions"

_safety, 2022_

Round 1

Reviewer 1 Report

The subject of the paper is very interesting and relevant. Problems associated with the use of robots and, first of all, security issues should be intensively investigated. This problem is studied in the paper.
This paper proposes an expert analysis of this problem. It is important. And this work can be published. However, the effectiveness and benefits of the proposed model should be presented in more detail and visually.

Reviewer 2 Report

The paper describes the creation of a System Dynamics model for a safe human robot interaction using results of a Delphi study. Practical aspects of applying such a model e.g. for designing a HRI system are not provided.

The introduction could highlight some existing technical realizations of HRI in order to show disregarded aspects, which are necessary for e.g. a safe and efficient interaction. 

The main outcomes are summarized in fig. 2 which visualizes the generated model. Being the main contribution of the paper, the figure should be much clearer. Some of the relations are hardly readable and the arrowheads are missing or invisible. In the current form the figure is quite difficult to understand. Please check as well the use of "optimalization". Optimization could be better.

Fig. 2 - and so the complete model - seems quite incomplete, considering existing collaborating systems. The "situational awareness" is nowadays a task of AI-based methods in order to predict the upcoming situation. This is one of the main contributions to safety. Here it is only associated with a human cluster.

In order to evaluate the generated model, it should be applied to a practical task e.g. a workspace monitoring system for mobile robots. This could help to validate the authors approach. 

In "§4 Discussion" eleven clusters are mentioned, whereas fig. 2 just shows ten. The same for the relationships. Please check the amount.

Line 168: "[...] Dutch normalization an standardization [...]" Is it an "and"?

Line 305: "[...] why was chosen for a system [...]" Sounds strange to me - please check grammar.

Reviewer 3 Report

The manuscript discussed the safety of cobot in complex human-robot shared scenarios, and presented a interaction SD model that has aggregated relevant factors to a tractable level,which can be used as a supportive tool for discussing cobot safety of machines and humans on a shared space. Some suggestions for your reference 1) Modify the format and layout of the chart in the paper 2) Unified reference format 3) Refine the abstract and titte for more attractive

Round 2

Reviewer 2 Report

There are still misleading information in fig. 2 after revision. Even if the underlying technology for e.g. situational awareness is not explicitly mentioned, the arrow from "Human vigilance" to "Safety" implies that the human co-worker is part of the perception chain. This can be nowadays as well part of the machine reliability but there is no arrow connecting these clusters.

The complexity of Fig. 2, in visualising the interactions between the identified clusters, should be quite low, but the authors should think about adding or changing connections btw. the main clusters - especially considering the technologies available nowadays. Some technologies even replace or move the interaction to different clusters allowing the human co-worker e.g. to reduce his "vigilance".

As already mentioned, the main output of the paper is fig. 2 and should cover the majority of HRI scenarios considering as well technological aspects. If there is no possibility of adding these aspects to fig. 2, the impact of current technologies to the identified clusters should be clearly mentioned in the discussion. Lines 426-435 do not cover these aspects sufficiently.
